# Association between Primary Healthcare and Medical Expenditures in a Context of Hospital-Oriented Healthcare System in China: A National Panel Dataset, 2012–2016

**DOI:** 10.3390/ijerph17186917

**Published:** 2020-09-22

**Authors:** Mengping Zhou, Jingyi Liao, Nan Hu, Li Kuang

**Affiliations:** 1Department of Health Administration, School of Public Health, Sun Yat-Sen University, Guangzhou 510080, China; zhoump@mail2.sysu.edu.cn (M.Z.); liaojy33@mail2.sysu.edu.cn (J.L.); 2Department of Biostatistics, FIU Robert Stempel College of Public Health and Social Work, Miami, FL 33199, USA; 3Department of Family and Preventive Medicine, and Population Health Sciences, University of Utah School of Medicine, Salt Lake City, UT 84132, USA

**Keywords:** primary healthcare, medical expenditures, ecological study, panel data

## Abstract

Total health expenditure in China has grown considerably since a new round of health system reform was enacted in 2009. Researchers have shown that strengthening primary healthcare may be an option for countries to solve the rapid expansion of their medical expenditures. This study was designed to explore the association between the strength of primary healthcare and medical expenditures, in the context of the hospital-oriented healthcare system in China. A longitudinal ecological study was conducted using a 5-year panel dataset of 27 provinces in mainland China. The linear mixed effects regression model was used to assess the effects of primary healthcare-related metrics on medical expenditures, controlling for the provincial level specialty care physician supply and socio-economic parameters. All of the three primary healthcare-related metrics showed negative associations with the two medical expenditure parameters. Primary care physicians per 10,000 population was significantly associated with the per capita hospital medical expenditures (*p* < 0.05), and the percentage of public health expenditure in total health expenditure was significantly associated with both per capita total medical expenditure and per capita hospital medical expenditures (*p* < 0.01 for both). Our study found negative associations between the primary healthcare capacity and medical expenditure in the context of hospital-oriented healthcare systems in China, adding to the previous evidence that primary healthcare may play a positive role in reducing medical expenditure. Policies on increasing the primary care physician supply and the public share of total health expenditure should be conducted to strengthen the primary healthcare system. With the gradual advance of medical reform and the policy inclination to primary healthcare, this will play a more important role in controlling the rapid growth of medical expenditure.

## 1. Introduction

To maintain the sustainable growth of medical expenditure has become an important issue facing the health system reform in all countries since the health sector continues to expand faster than the economy [1]. Between 2000 and 2017, global health spending in real terms grew by 3.9% a year while the economy grew by 3.0% a year and this trend was even more serious in middle-income countries (expenditure 6.3% vs. GDP 5.9%) [2]. As one of the middle-income countries, China is also rapidly converging towards higher levels of spending since the new round of health system reforms was enacted in 2009 [3]. Total health expenditure in China grew at a rate of 14.45% per annum in 2009–2018, which is higher than that of GDP (12.89%), leading to an increase in the health expenditure share of GDP from 4.96% to 6.39% during the same period [4,5]. With the disease spectrum changing, population aging, medicinal and technological developments, and people’s health needs increasing, this problem will get worse in China [3]. According to one study that projected health expenditure in the period 2015–35 in China, the health expenditure is expected to rise to 9.1% of GDP in 2035 [6]. This rapid growth in health expenditure imposes pressures on the government budget as well as on the personal finances of patients, especially in the light of the slowdown in economic growth [7]. Without optimizing the structure of the health system and integrating health services, increased investment would not be transferred to improved access and the rising medical cost will not be effectively controlled, and thus the goal to implement UHC would be jeopardized [8]. In recent years, healthcare systems have been dedicated to better organizing health services to maximize both care effectiveness and efficiency [9].

Primary healthcare (PHC) is recognized as the most cost-effective investment to solve the problem of increased medical expenditure [10,11,12]. The Declaration of Alma-Ata in 1978, followed by the 2008 World Health Report and the Declaration of Astana in 2018, all defined PHC as the foundation of an effective health system and emphasized that all countries should strengthen the PHC system to overcome the emerging challenges [13,14,15]. PHC has been widely considered to be an effective strategy to realize universal health coverage, achieve the health-related Sustainable Development Goals, and improve health outcomes by emphasizing disease prevention and health promotion, ensuring equity of access to the most essential interventions [16,17]. Researchers all over the world have shown that the PHC system plays an important role in achieving the effective control of medical costs. Many PHC-related programs have been confirmed to be cost-effective and sustainable [18]. Health systems with a PHC orientation or stronger PHC structure can reduce healthcare spending without lowering the quality of care [19,20,21]. Universal PHC is helpful in guiding the reasonable needs of patients, reducing the waste of resources, such as specialists in large hospitals dealing with simple diseases, and relieving the challenges of difficult and costly access to hospitals [22]. There is now an international consensus that all countries should enhance PHC to ensure the containment of rising medical expenditure [23].

China has responded positively to international appeals to strengthen the PHC system. As the first tier of the healthcare system in China, the PHC system provides generalist clinical care and basic public health services [24]. In the current decade, the Chinese government has enacted a number of new policies and health reforms to build a stronger PHC system in terms of building infrastructure, training personnel, providing public health services, and developing other supporting programs [22,25]. The government increased its subsidies to primary healthcare institutions from CNY 19.8 billion (USD 2.8 billion) in 2008 to CNY 197.7 billion (USD 27.8 billion) in 2018 [26]. In addition, the government has significantly increased public health expenditure during the same period, with the public share of total health expenditure rising from 59.6% to 71.3% [4]. Adequate and equitable funding is crucial to protecting families with financial hardship from losing primary health services [20]. The implementation of the new “5 + 3” generalist training programs (5 years of undergraduate clinical medicine + 3 years of standardized training) and relevant incentives has led to an improved professional ability and an increase in the salary of primary healthcare workers [27]. All Chinese citizens have been entitled to 14 free public health services, which are provided by primary healthcare institutions and the subsidy for these services has risen from CNY 15 in 2009 to CNY 69 per capita in 2019 [28]. Despite these facts, people in China still heavily depend on hospital-based healthcare. In 2018, 43.1% of doctor’s visits were in hospitals [26], even if the primary healthcare system provided higher geographical coverage and economic affordability than hospitals. The average cost of an outpatient visit to a primary healthcare institution was about CNY 101.9 (USD 14.4), including the cost of prescribed drugs, compared to CNY 274.1 (USD 38.8) per hospital outpatient appointment in 2018 [4].

Accumulating researches have been exploring the positive effects of PHC on reducing medical expenditures. Margatet et al. [20] reviewed plenty of existing studies to assess the contributions of major PHC-focused initiatives implemented in low- and middle-income countries and concluded that these initiatives have positive effects on impoverishing health spending by increased access to services and more rational drug use, as well as reducing the financial burden on families by reducing over-treatment and the excessive use of medication. A study in Greece showed that the lack of an integrated PHC system such as gatekeeping mechanisms, patient monitoring, and general population prevention programs may be the reason for the significant and consistent increases in both public and private expenditure [29]. But both conclusions above were just conjecture and the researchers did not confirm a quantitative association between PHC system and medical expenditure. Besides, existing evidence includes reports from developed and developing countries, but there are few studies from China. Addressing the association in China is essential to understand whether efforts to strengthen the PHC system have the potential to curb increasing health spending. As the rapid expansion of medical expenditure has become an important public health issue facing global health system reform, quantitative results from China may also provide evidence to support the expansion of the PHC system in countries all over the world.

Our study uses provincial-level PHC-related data from 2012–2016 to investigate whether the PHC can effectively reduce the medical expenditures in China. We hope this study will serve as an important reference in this area and provide some policy implications for global health system reform.

## 2. Materials and Methods 

### 2.1. Study Design

A longitudinal ecological design was used for this study. We used the provincial-level panel data from secondary sources to evaluate the effect of PHC-related metrics on selected medical expenditures using multi-variable models that control for the covariates of specialty care physician supplies and provincial level socio-economic status. The basic analysis units were the 27 provinces in mainland China (four province-level administrative municipalities, Beijing, Shanghai, Tianjin, and Chongqing were excluded from the analysis due to the incomparable medical expenditures). Data were collected during 2012–2016, and the number of observations was 135 (27 provinces × 5 years).

### 2.2. Data Resource and Variables

The medical expenditure indicators and the proportion of public health expenditure was excerpted from the China National Health Accounts Report 2013–2017. The rest of the variables were all obtained from the China Statistical Yearbook on Health and Family Planning 2013–2017 and the National Bureau of Statistics. 

#### 2.2.1. Dependent Variables

The dependent variables are per capita total medical expenditures and per capita hospital medical expenditures. The total medical expenditures refer to the total health expenditure after deducting the expenses of drug institutions, public health institutions, administrative agencies, and other health institutions. The hospital medical expenditure refers to total health expenses flowing into hospital institutions. 

#### 2.2.2. Exposures

Exposures are those related to the strength of PHC. Three inter-related and synergistic components of PHC were identified according to the WHO report: primary care and essential public health functions as the core of integrated health services, multisectoral policies and action, empowered people and communities [16]. Due to the data availability, we used three variables to represent two important components of PHC. The variable of primary care physician supply was defined as the number of primary care physicians per 10,000 population at each province, which is a parameter of primary care and essential public health functions of PHC. Primary care physicians refer to all licensed doctors and doctor assistants who work in primary health institutions, which includes community health centers, township health centers, outpatient clinics and village clinics according to National Health Statistics Center of China. Both the popularity rate of sanitary toilets in rural areas and the proportion of public health expenditure in total health expenditure were used as the parameters for multisectoral policies and action domain of PHC. The sanitation improvement is an important goal of the famous Patriotic Health Campaign in China, which is a whole-of-government approach to multisectoral policy and action at the national, subnational, and regional levels [30]. The proportion of public health expenditure was used to capture the expected positive role of government in determining healthcare expenditure [31], defined as the sum proportion of government health expenditure and social health expenditure. 

#### 2.2.3. Covariates

Factors such as health systems characteristics, income, demographic structure, education, and occupation have been commonly stated in international literature as influencing the level and growth of health expenditures [32]. Hence, we included the number of hospital-based specialty care physicians per 10,000 population, GDP per capital, the proportion of the population aged ≥65, the proportion of illiterate population aged ≥15, and the registered urban unemployment rate as covariates. Inclusion of a per capita income variable is standard practice in studies of healthcare expenditures determinants [33]. 

### 2.3. Statistical Analysis

A linear mixed effects regression model for panel data was used to assess the associations between PHC-related metrics and two medical expenditure parameters while controlling for the confounding factors. We hypothesized the exchangeable correlation structure among the repeated expenditures within a province, allowing intercepts to vary among provinces. Independent variables, including three PHC-related indicators and other potential confounders, were treated as fixed effects. Year was also included in the model as a fixed effect to control for unmeasured time-variant characteristics of all provinces. The equation for the mixed effects regression model can be written as:*Y_it_* = *β*_0_ + *β**X**_it_* + *δT_t_* + *α_i_* + *ε_it_*(1)
where *i* (*I* = 1⋯, 27) represents the province, *t* is the number of years since 2012 (*t* = 0⋯,5), *Y_it_* represents the medical expenditure variable for province *i* at time *t*, ***X****_it_* denotes the vector of independent variables for province *i* at time *t*, *T_t_* is a vector of year-dummies (have *t*−1 time periods). *β*_0_ is the mean intercept of the 27 provinces over the 5 years, *β* is the vector of regression coefficients of independent variables, *δ* is the vector of coefficients for the year dummy variables, *α_i_* is assumed as the province-specific random effect, and *ε_it_* is the random error term.

All analyses were conducted using Stata (StataCorp., College Station, TX, USA) version 14 and Figure 1 was plotted using the statistical package R (www.r-project.org) version 3.6.1. All tests are two-sided; *p*-values < 0.05 indicate statistically significant results. Robust standard errors (SEs) were used for the analysis of the panel data.

## 3. Results

Table 1 shows descriptive statistics results of all variables from 2012 to 2016. Both per capita total medical expenditures and per capita hospital medical expenditures per capita grew rapidly during this period, from about CNY 1390 to CNY 2220, and from 1070 to CNY 1740, respectively. As for the three indicators of PHC, they all increased smoothly from 2012 to 2016, with the number of primary care physicians per 10,000 population increasing from 7.15 to 7.90. The proportion of public health expenditure among total health expenditure increased from 65.51% to 71.34%, and the popularity rate of sanitary toilets in rural areas rose from 68.83% to 77.82%. During the same period, the number of hospital-based specialty care physicians per 10,000 population also showed a steady increase, from 10.29 in 2012 to 12.95 in 2016.

Table 2 presents the results of mixed effects linear regression models. All of the three PHC-related metrics showed a negative association with per capita total medical expenditures and per capita hospital medical expenditures, among which primary care physicians per 10,000 population was statistically significant in the hospital medical expenditure model (*p* < 0.05) and the proportion of public health expenditure was statistically significant in both models (*p* < 0.01). Take the proportion of public health expenditure for example, a per-unit increase in the proportion of public health expenditure was associated with a CNY 33.1 (USD 4.68) decrease in per capita total medical expenditures and a CNY 26.4 (USD 3.74) decrease in per capita hospital medical expenditures, while other variables were held constant. In order to make the relationship between PHC-related metrics and medical expenditure parameters more visual, we drew six plots of estimated medical expenditure versus three primary healthcare-related metrics (Figure 1).

Specialty care physicians per 10,000 population and GDP per capital were both positively and significantly associated with both per capita total medical expenditures and per capita hospital medical expenditures. The goodness-of-fit results showed that the R2 of both models was over 93%, which meant that the regression models capture 93% of all variations of observed dependent variables and covariates. All independent variables used in the models were checked for multicollinearity, which was not found (results not shown).

In each plot, the other two primary healthcare-related metrics and all covariates (specialty care physicians per 10,000 population, GDP, the proportion of the population aged ≥ 65, the proportion of illiterate population aged 15 and above, and registered urban unemployment rate) were set as the mean values.

## 4. Discussion

Few studies have investigated the association of PHC with medical expenditure in China. This study was intended to fill the gap by conducting an ecological study using a province-level panel dataset of China. We found a negative and significant association between PHC-related metrics and medical expenditure. These findings suggest that PHC may have positive effects on reducing health spending, which is especially meaningful in China since the contribution of PHC is neglected and hospital-based healthcare still dominates the health system.

The three PHC-related metrics were all negatively associated with both per capita total medical expenditure and per capita hospital medical expenditure, and the proportion of public health expenditure as well as the primary care physician supply were statistically significant either in one model or both models after controlling for the specialty care physician supply and socio-economic status, confirming that PHC might play an overall positive role in reducing health spending in China. This result is consistent with the findings within countries such as the United States [34], the United Kingdom [35] and other low- and middle-income countries [36]. International comparisons also showed that those countries with weaker PHC systems had significantly higher costs [37,38]. Previous studies have explained that PHC may work through several mechanisms. From a macroeconomic perspective, PHC plays a key role in reducing health expenditure by addressing the underlying determinants of health, by emphasizing population-level services and by improving health equity [11,21]. This both reduces the need for individual care and can avoid the escalation of health issues to more complex and costly conditions. From a microeconomic perspective, the core functions of PHC, such as continuity, comprehensiveness, accessibility, coordination of care, and community-based services, play an important role in reducing wasteful use of healthcare resources (including avoidable hospitalizations, readmissions to hospital, additional specialty referrals, laboratory and diagnostic tests, and unnecessary use of emergency departments), thus reducing healthcare costs [39,40,41,42,43].

Our study showed that the proportion of public health expenditure plays an important role in reducing the per capita total medical expenditures and hospital medical expenditures. Study showed that public health expenditure was an important factor of public participation in healthcare financing, which reflected how much attention the government paid to improve the public access to healthcare services among developing countries [44]. In China, public health expenditure consists of both government and social investment in healthcare enterprises, among which government funds are used for the construction of basic medical facilities and the provision of Basic Public Health Services, and the social health insurance funds enable more than 95% of the population in China to have basic health insurance. Public health expenditure plays an important role in reaching some millennium development goals such as achieving universal health coverage and improving health outcomes. Studies in Sub-Saharan Africa, EMR countries and OECD countries all confirmed that a higher proportion of public health expenditure over total health expenditure had a strong positive relationship in terms of health status [45,46,47]. With the improvement in health status, the rising medical cost has been controlled.

As the backbone of the primary healthcare workforce, primary care physicians play a vital role in controlling medical expenses and guiding patients in terms of reasonable health needs [22]. Our study confirmed that, in the context of China, there is a significant negative correlation between the primary care physician supply and per capita hospital medical expenditure. Several studies in the United States also found that a higher proportion of primary care physicians in an area was associated with a lower level of spending (measured by per beneficiary Medicare spending and Medicare end-of-life spending) [21,34,48,49]. Evidence shows that, compared with specialists, primary care physicians lead to savings in public medical resources, such as hospitalizations, prescriptions, common tests, and procedures [49,50]. They also provide more comprehensive health services, better preventive care, and early management of health problems [21], which may also lead to a reduction in medical expenditure.

Our study showed a negative association between the popularity rate of sanitary toilets in rural areas and two medical expenditure outcomes but the correlation was statistically insignificant. As an important part of the Patriotic Health Campaign and the Healthy Village Construction in China, the sanitary toilet revolution is committed to improving rural sanitary conditions and human settlements and the popularity rate of rural sanitary toilets is an important indicator to measure the effect of revolution. After years of unremitting effort, the rebuilding of rural toilets has demonstrated remarkable achievements, with the popularity rate of rural sanitary toilets increasing from 40.3% in 2000 to 81.8% in 2017 [51]. These sanitary toilets have met the basic standards of enclosed, ventilated, sealed, and covered septic tanks, with no fly maggots or persistent odors. The obvious improvement of basic sanitary conditions in rural areas effectively controls the occurrence and prevalence of diseases, obviously improving the quality of civilization and health literacy of peasants, and significantly promoting the improvement of the rural ecological environment, so that health, ecological, economic, and social benefits gradually emerged [52]. However, our study did not find a significant association between the provincial level popularity rate of sanitary toilets and medical expenditure, possibly because the panel data was too short to reflect the rapid increase in the popularity rate of sanitary toilets around 2015.

Notably, specialty care physicians per 10,000 population was found to be significantly and positively associated with both per capita total medical expenditure and per capita hospital medical expenditure and the regression coefficient was larger than that of each PHC-related metric, implying that specialty care physicians may have an inescapable responsibility for the growth of medical expenditure. This result was consistent with the research in US which found that states with a higher proportion of specialists have a higher cost per beneficiary [34]. The more expensive medical resources and overuse of tests and diagnostics may be responsible for this association.

Our study showed a strong positive relationship between GDP and medical expenditure, which is highly consistent with previous research [33,53]. GDP was identified as a significant statistical factor having positive influence on national healthcare expenditure increase [32]. Previous studies have also demonstrated that aging populations have a significant positive influence on total health expenditure [54]. Our study did discover a positive relationship between the proportion of the population aged ≥ 65 and medical expenditure, but the association was insignificant. Most literature showed that GDP and aging populations were two key explanatory variables for the growth of health expenditure [33,55]. Our consistent findings with existing studies showed the reliability of the results. After controlling for the two major confounders, our finding shows a real association between PHC-related metrics and medical expenditure.

Our study demonstrates that health systems which depend heavily on PHC are more advantageous than those heavily based on specialist care in terms of reduction in medical expenditure. Findings support policies that encourage a shift of services away from specialist care to PHC. In order to maximize the potential of PHC on reducing medical expenditure, national efforts should be directed at enhancing the primary care workforce as well as increasing the public share of total health expenditure.

### Study Limitations

This research has several limitations. First of all, due to the nature of this ecological study, it is unable to establish the relationship between the subject level characteristics and the out-of-pocket medical expenditure for individuals. Second, it is possible there are latent and unmeasured provincial-level variables that may confound the relationship between medical expenditure and PHC-related parameters. Endogeneity may exist due to these omitted variables or simultaneity. Lastly, parameters related to the other component of PHC (namely empowered people and communities) were not available in this dataset, so we can only use three parameters to represent the impact of PHC on medical expenditure. Future studies should use more representative and comprehensive indicators to measure the strength and quality of PHC.

## 5. Conclusions

To our knowledge, this is one of the few studies that have explored the association of PHC with medical expenditure using national panel data in China. As the medical expenditure varies widely by province in China, our province-level study constitutes an important case study and provides policy implications for low-, middle-, and high-income countries. The study findings showed that the strength of PHC was negatively associated with medical expenditure, adding new evidence of the potential salutary impact of PHC on reducing health spending and providing good policy implications for countries all over the world. To exert more influence in controlling the growth of medical expenditure, the global health system reform should continue to strengthen the PHC system by increasing the primary care physician supply and the public share of total health expenditure and encouraging a shift of services away from specialist care to PHC.

## Figures and Tables

**Figure 1 ijerph-17-06917-f001:**
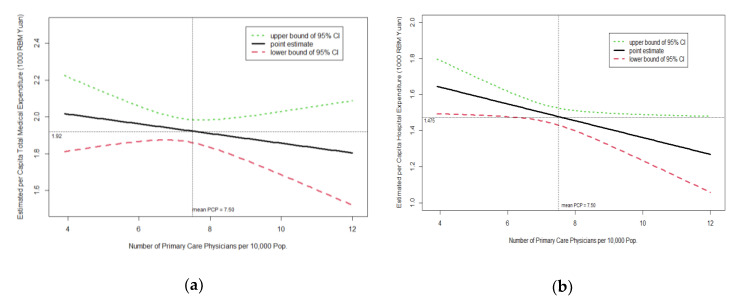
Estimated medical expenditures and 95% confidence intervals (CIs) versus three predictor variables. The estimation is based on a linear mixed-effects model. (**a**) Estimated total expenditure versus primary care physicians supply in year 2015; (**b**) estimated hospital medical expenditure versus primary care physicians supply in year 2015; (**c**) estimated total expenditure versus proportion of public health expenditure in year 2015; (**d**) estimated hospital medical expenditure versus proportion of public health expenditure in year 2015; (**e**) estimated total expenditure versus popularity rate of sanitary toilets in rural areas in year 2015; (**f**) estimated hospital medical expenditure versus popularity rate of sanitary toilets in rural areas in year 2015.

**Table 1 ijerph-17-06917-t001:** Descriptive statistics among the 27 provinces in China, 2012–2016.

Variable	2012		2013		2014		2015		2016	
	Mean	SD	Mean	SD	Mean	SD	Mean	SD	Mean	SD
Per capita total medical expenditures (1000 Yuan)	1.39	0.28	1.58	0.33	1.74	0.38	1.96	0.43	2.22	0.48
Per capita hospital medical expenditures (1000 Yuan)	1.07	0.26	1.22	0.30	1.35	0.33	1.52	0.37	1.74	0.41
Primary care physicians per 10,000 population	7.15	1.38	7.38	1.35	7.44	1.33	7.66	1.40	7.90	1.45
The proportion of public health expenditure, %	65.51	7.51	66.49	7.36	68.25	7.00	70.34	6.39	71.34	6.05
Popularity rate of sanitary toilets in rural areas, %	68.83	13.63	71.46	13.74	73.27	13.76	76.21	12.32	77.82	11.74
Specialty care physicians per 10,000 population	10.29	2.01	11.04	2.01	11.53	1.98	12.27	1.97	12.95	2.04
GDP per capital (10,000 Yuan)	3.85	1.37	4.20	1.45	4.53	1.55	4.73	1.64	4.97	1.72
Proportion of the population aged ≥65, %	8.76	1.60	9.01	1.67	9.35	1.85	9.78	1.87	10.06	2.10
Proportion of illiterate population aged 15 and above, %	17.24	3.63	17.14	3.48	17.18	3.58	17.31	3.73	17.33	3.67
Registered urban unemployment rate, %	3.40	0.54	3.35	0.57	3.30	0.56	3.28	0.59	3.27	0.59

**Table 2 ijerph-17-06917-t002:** Regression model results for the 27 provinces of China, 2012–2016.

Variable	Per Capita Total Medical Expenditure (1000 Yuan)	Per Capita Hospital Medical Expenditure (1000 Yuan)
Primary care physicians per 10,000 population	−0.026 (−0.088 to 0.035)	−0.046 * (−0.092 to −0.001)
The proportion of public health expenditure, %	−0.033 *** (−0.049 to −0.018)	−0.026 ** (−0.039 to −0.014)
Popularity rate of sanitary toilets in rural areas, %	−0.004 (−0.012 to 0.005)	−0.003 (−0.009 to 0.003)
Specialty care physicians per 10,000 population	0.103 * (0.013 to 0.193)	0.102 * (0.027 to 0.176)
GDP per capital (10,000 Yuan)	0.132 * (0.019 to 0.245)	0.118 * (0.014 to 0.221)
Proportion of the population aged ≥65, %	0.025 (−0.027 to 0.077)	0.020 (−0.024 to 0.065)
Proportion of illiterate population aged 15 and above, %	0.001 (−0.027 to 0.023)	−0.001 (−0.015 to 0.014)
Registered urban unemployment rate, %	−0.008 (−0.138 to 0.123)	0.003 (−0.100 to 0.106)
Constant	2.855 * (1.323 to 4.388)	2.109 (0.995 to 3.222)
Observations	135	135
Number of provinces	27	27
R-squared (within)	0.939	0.931

* *p* < 0.05, ** *p* < 0.01, *** *p* < 0.001. Year fixed effects not shown; 95% CI was in the parentheses.

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
