# Peer review of "Association between Primary Healthcare and Medical Expenditures in a Context of Hospital-Oriented Healthcare System in China: A National Panel Dataset, 2012–2016"

_ijerph, 2020, doi:10.3390/ijerph17186917_

Round 1

Reviewer 1 Report

The study conducted by Zhou, et al. aimed to explore the association between the strength of primary health care and medical expenditures using a 5-year panel dataset of 27 provinces in mainland China.

The Introduction section provided sufficient background for the study rationale. The research method was appropriate and adequately described. The conclusion on negative associations between primary health care capacity and medical expenditures were well supported by the results.

One may speculate that the rise in price level during 2012-2016 might play a role in the escalation of total medical expenditures from CNY1390 (2012) to CNY2220 (2016) and hospital medical expenditures from CNY1070 (2012) to CNY1740 (2016). Thus the association observed between the capacity of primary health care and medical expenditures might be somewhat attenuated by price inflation in the context of China's rapid economic development. This may be worthy of mention in the discussions on limitations of the study.

Overall, the manuscript was nicely written and can be considered for publication at IJERPH.

Author Response

Dear Reviewer:

We greatly appreciate your positive and insightful review to help us improve the manuscript. In the study design, we considered the possible effect of price level on medical expenditures, but we did not include this indicator due to the collinearity with GDP. Besides, our statistical model has included year as a fixed effect to control for unmeasured time-variant characteristics of all provinces. So, the price inflation may be the one that the model controlled. But there is still a possibility that the association observed may be somewhat attenuated. So, based on your suggestion, we have explained this in the limitation part.

Reviewer 2 Report

Review of “Association Between Primary Health Care and Medical Expenditures in a Context of Hospital-Oriented Healthcare System in China: A National Panel Dataset, 2012–2016

The paper aims to investigate whether the primary healthcare can effectively reduce the medical expenditures. The results are based on the data about different Chinese provinces. The research is interesting and well done, showing some relevant relationship in this interesting and debated topic. However, the manuscript needs some improvements in my opinion.

Below, the main points are described:

  • The first point concerns the perspective given to the paper, in particular in the introduction and conclusion. I think the paper should stress better the international level of the research. The manuscript should highlight the general contributions concerning your research and results, obtained through a case study carried out in China (a "middle-income" country). In my opinion, China constitutes an important and interesting case study, but the research the results should be discussed in a more international point of view. A more international perspective is therefore highly recommended.

  • The second point is related to the background. The authors should reinforce the background of the paper by enlarging the introduction or creating a new dedicating section “background”. Hereafter a list of useful papers which advise is provided: - Panesar, S. S., Desilva, D., Carson-Stevens, A., Cresswell, K. M., Salvilla, S. A., Slight, S. P., ... & Bates, D. W. (2016). How safe is primary care? A systematic review. BMJ quality & safety, 25(7), 544-553. - Basu, S., Berkowitz, S. A., Phillips, R. L., Bitton, A., Landon, B. E., & Phillips, R. S. (2019). Association of primary care physician supply with population mortality in the United States, 2005-2015. JAMA internal medicine, 179(4), 506-514.  - Stefanini, A., Aloini, D., Benevento, E., Dulmin, R., & Mininno, V. (2020). A data-driven methodology for supporting resource planning of health services. Socio-Economic Planning Sciences, 70, 100744. https://doi.org/10.1016/j.seps.2019.100744    - Fairall, L., Bateman, E., Cornick, R., Faris, G., Timmerman, V., Folb, N., ... & Smith, R. (2015). Innovating to improve primary care in less developed countries: towards a global model. BMJ innovations, 1(4), 196-203.    - Lau, R., Stevenson, F., Ong, B. N., Dziedzic, K., Treweek, S., Eldridge, S., ... & Peacock, R. (2015). Achieving change in primary care—causes of the evidence to practice gap: systematic reviews of reviews. Implementation Science, 11(1), 40.

  • The results are interesting. But, I think some more explanations about the statistical model building can help the reader to understand your analysis process. Just for example, it is not clearly expressed how you have 135 observations: I understood that is 27 (province) * 5 (years), but it is better to clearly state it. I think it is important to in-depth explain your statistical models and results obtained.

  • I think the limitations can be reported in the conclusion rather than in the discussion or in a sub-section of the discussion section. Otherwise, the research limitations are “confused” with the rest of the discussions of your results in light of past literature.

Reviewer 3 Report

REFEREE REPORT

Manuscript ID: ijerph-921434

Title: Association between primary health care and medical expenditures in a context of hospital-oriented healthcare system in China: a national panel dataset, 2012-2016

Summary

The paper examines whether primary health care lowers medical expenditures in China.  The dependent variables are per capita total medical expenditures and per capita hospital medical expenditures. The explanatory variables that capture access to primary care are number of primary care physicians, proportion of public health expenditures and popularity rates of sanitary toilets in rural areas. The findings show that some measures of access to primary health care are significantly negatively associated with medical expenditures.

The paper is highly relevant and interesting in the field of health economics. I have a few questions and suggestions.

Comments/ Questions:

The linear mixed effects model was not described at all. What do you mean by mixed effects model?

Potential for endogeneity as not addressed – endogeneity either from simultaneity or omitted variable problem. For example, it is possible that there may be confounding unobserved factors that affected both number of primary care physicians (explanatory variable) at each province that also affected expenditures in hospitals in that area (dependent variable).  This can effect consistency of the estimates.

The result illustrated by the figures show that the effect of each explanatory variable on the dependent variable varies by the level of the explanatory variables. But the model in (1) is an OLS and linear in all variables. Shouldn’t the marginal effects just be constant?  I don’t see any quadratic or interaction terms. Please explain.

Are the monetary estimates economically relevant? How much are those $ estimates of marginal effects relative to mean per capita income for the year? If they are too small, the economic relevance of the result may be small even if they are statistically significant.

The paper and its results would have had more bite if it was a true treatment effects model such that the dataset includes pre and post 2009, when the health system reform was enacted. Instead, it only covers 2012-2016.  It is unfortunate that the results imply association only and cannot be interpreted more strongly. 

Round 2

Reviewer 2 Report

The authors have improved the paper following the instructions provided.